# Evolution of the Growth Mode and Its Consequences during Bulk Crystallization of GaN

**DOI:** 10.3390/ma16093360

**Published:** 2023-04-25

**Authors:** Tomasz Sochacki, Robert Kucharski, Karolina Grabianska, Jan L. Weyher, Magdalena A. Zajac, Malgorzata Iwinska, Lutz Kirste, Michal Bockowski

**Affiliations:** 1Institute of High Pressure Physics, Polish Academy of Sciences, Sokolowska 29/37, 01-142 Warsaw, Poland; 2Faculty of New Technologies and Chemistry, Military University of Technology, Kaliskiego 2, 00-908 Warsaw, Poland; 3Fraunhofer Institute for Applied Solid State Physics (IAF), Tullastraße 72, 79108 Freiburg, Germany

**Keywords:** GaN, crystal growth, basic ammonothermal method, halide vapor phase epitaxy, growth morphology, growth model

## Abstract

A detailed analysis of morphology of gallium nitride crystal growth obtained by ammonothermal and halide vapor phase epitaxy methods was carried out. The work was conducted to determine the source of triangular planar defects visible in X-ray topography as areas with locally different lattice parameters. It is shown that the occurrence of these defects is related to growth hillocks. Particular attention was paid to analyzing the manner and consequences of merging hillocks. In the course of the study, the nature of the mentioned defects and the cause of their formation were determined. It was established that the appearance of the defects depends on the angle formed between the steps located on the sides of two adjacent hillocks. A universal growth model is presented to explain the cause of heterogeneity during the merging of growth hillocks.

## 1. Introduction

Gallium nitride (GaN) substrates of high structural perfection and homogeneous electrical properties are required for advanced electronic and optoelectronic devices based on GaN-on-GaN technology [1,2]. Currently, bulk GaN is crystallized by three methods: from the gas phase by Halide Vapor Phase Epitaxy (HVPE) and from solution by sodium flux as well as ammonothermal methods [3,4,5,6]. The key to obtaining crystals of high structural quality is using a proper seed, allowing for the best match of lattice parameters with the grown layer, and conducting stable growth without disturbances on the crystallization front [7]. Regardless of the method, bulk crystallization in polar [000$\overset{-}{1}$] or [0001] (hereafter designated as −c and c, respectively) directions on seeds with a given misorientation can be realized in a similar way in terms of successive changes in growth modes. In most of the observed cases, the growth begins in a step propagation mode resulting from the substrate off-cut. Next, a low-energy polar plane was reconstructed [8,9]. On top of it, hexagonal hillocks form, and they become the new source of the propagation of steps. If several or more hillocks are formed, they can merge under proper conditions. This leads, in the best scenario, to the formation of a single hillock on which bulk growth is finally realized. Crystallization on a single hillock seems to be the most optimal way for bulk GaN. An analysis of growth modes and transitions between them seems to be crucial for understanding the crystallization process. Despite tremendous progress in the field of GaN crystallization, such studies need to be further conducted, especially for the ammonothermal method due to a difficulty in direct analysis of growth during the process.

It was reported that crystals of the highest structural quality were obtained by the basic ammonothermal method [10]. In addition, when ammonothermal GaN (Am-GaN) crystals are used as seeds, their quality can be reflected in the HVPE method [11,12]. The high crystallographic quality of Am-GaN in terms of high bowing radii of crystallographic planes and low dislocations density allowed for the observation of the Borrmann effect (anomalous transmission of X-rays) in X-ray topography (XRT) in transmission mode [13]. X-ray topography analysis enabled a comprehensive extraction of a wide range of different defects and inhomogeneities in the studied material [13,14]. However, it should be emphasized that the observation and isolation of individual structural imperfections of the substrate in the aforementioned measurement technique was possible due to the high structural quality of the GaN substrate.

The starting point for the following research was the need to find the causes of the newly revealed triangular planar defects observed by XRT in GaN substrates. Figure 1 gives an example of these defects. They were described by Kirste et al. as “traces of facet formation” [13]. The source and the nature of these defects have not been fully explained so far. 

This paper presents a detailed analysis of the growth morphology of a crystal obtained by the ammonothermal method. Particular emphasis was placed on investigating the mutual arrangement of the growth hillocks and its influence on the growth morphology and properties of the crystallized GaN layer. The conducted research allowed us to build a universal model of connection of the growth of hillocks. The model presented in this paper is a supplement to the general one of GaN crystal growth on substrates with a given off-cut. The universalism of the model was verified on the basis of the analysis of a GaN crystal obtained from the HVPE method.

## 2. Methods

For the purpose of this work, two GaN crystals with sizes of 2 inches and 1 inch were selected. They were grown by the basic ammonothermal and HVPE methods, respectively. Both crystals were grown on native substrates and were of high structural quality in terms of high radius of crystallographic planes (above 20 m) and low threading dislocation density (10^4^ cm^−2^). Details of a standard basic ammonothermal process as well as the crystal growth zone configuration, including attaching the seeds to metal holders, were described elsewhere [10,15,16,17]. The crystal after the growth process was etched in order to remove a polycrystalline GaN layer that was deposited on the surface during the autoclave cooling phase. 

The HVPE growth process parameters used are published in ref. [8]. It is important to note that the time of the HVPE process was set in such a way to obtain a crystal on which growth will still be carried out on multiple hillocks rather than on a single one. It should also be stressed that the analysis was performed on as-grown surfaces, which for the ammonothermal and HVPE methods correspond to −c- and c-surfaces, respectively.

The morphology of the crystals and their luminescence were analyzed by a Nikon Eclipse LV100ND optical microscope (OM) under VIS light with Nomarski contrast and under UV illumination, respectively. Next, the surface of the Am-GaN crystal was prepared to an epi-ready state and photo-etched (PE) in modified KSO-D solution (0.02 M K_2_S_2_O_8_ + 0.02 M KOH) [18,19] with an addition of a 0.02 M Na_3_PO_4_ component in order to increase the stability of the solution [20]. A galvanic mode was employed for revealing electrically active inhomogeneities. PE was performed under illumination of a 300 W UV-enhanced Xe lamp (Oriel, Germany). On selected areas, photoetching depth profiles were recorded using a Tencore Alfa step profiler.

Based on the research, a model describing the formation of triangular defects was built.

## 3. Results

Figure 2 shows the morphology of a 2-inch as-grown Am-GaN crystal. The [a] and [m] designations correspond to the $<\bar{1}2\bar{1}0>$ and $<10\bar{1}0>$ direction families, respectively. In the figure, four developed growth hillocks are marked as white hexagons. The side walls of the hexagons are parallel to the macro-steps located on the slopes (hereinafter also referred to as sides) of the growth hillock. Through the center of each hexagon, lines were drawn that coincide with the six edges of the growth hillock. On the slopes of all the analyzed hillocks, macro-steps were observed (see Figure 3a). Directions of macro-steps propagation were marked with white arrows in Figure 2.

Areas at the junction of two side walls of adjacent hillocks were also analyzed. In some of them, disturbances in the ways of step propagation, i.e., step-bunching, were found (see Figure 3b). In Figure 2, the red dotted line indicates areas where step-bunching was observed. In addition, a red solid line shows the macro-steps on the hillock side adjacent to the region in which the disturbance occurs. Step-bunching was not observed on all junctions between two adjacent growth hillocks. Junctions without growth disturbances are marked with a yellow dotted line in Figure 2. Macro-steps adjacent to these areas are marked with a yellow solid line.

An additional analysis was performed on regions at the junction of two different growth morphologies, presented in Figure 4a,c. The same areas were inspected with OM under UV light (see Figure 4b,d). The places where the macro-steps are visible correspond to a region of darker luminescence visible under UV light. In turn, areas with visible destabilization (step-bunching) correspond to parts with brighter luminescence.

Figure 5 shows the second analyzed crystal: 1-inch-GaN from the HVPE method. The surface after growth with numerous growth hillocks is visible.

Parts of the surface indicated with white rectangles in Figure 5 were chosen for the analysis of the HVPE-GaN crystal. Figure 6a shows the first selected area (rectangle A) with developed growth hillocks, indicated with white hexagons. Red solid and dotted lines mark, respectively, steps on the sides of hillocks and areas where step-bunching is observed. The same part of the crystal indicated by rectangle A and visible in Figure 6a is shown under UV illumination in Figure 6b. The areas with visibly brighter luminescence correspond to step-bunching in Figure 6a. Some disturbances in step propagation from the part of the crystal indicated by rectangle B (see Figure 5) are shown more clearly in Figure 6c. Figure 6d shows the same part of the crystal as presented in Figure 6c but observed under UV light.

Next, the −c- and c-plane surfaces of the Am-GaN and HVPE-GaN crystals were prepared to an epi-ready state by mechanical and chemo-mechanical polishing and cleaning. Photo etching was carried out in the areas shown in Figure 4c,d (Am-GaN) and Figure 6c,d (HVPE-GaN). The photo-etched surfaces of the crystals are shown in Figure 7a,c. The shapes of the dark, deeper-etched areas correspond to the shapes of the parts with brighter luminescence visible in UV light shown in Figure 6d. In addition, in Figure 7a,c segments AB and CD were indicated along the profilometric measurement of the depth that was made (the profiles are presented in Figure 7b,d). The measurement showed that the darker areas are about 400 nm and 300 nm deeper with reference to the lighter part of the surfaces seen in Figure 7a,c, respectively.

## 4. Discussion

An analysis of the growth morphology of the GaN crystals obtained by the ammonothermal and HVPE methods allowed us to determine the cause of the triangular defects visible both under UV light and in the X-ray topography in transmission geometry. The triangular planar defects defined by Kirste et al. as “traces of facet formation” show up in X-ray topography as areas with different contrast due to a slight bending of the reflecting planes along the boundaries [13]. This is because there is a minor difference in lattice parameters between adjacent regions. Kirste et al. were right to assume that it could be related to a different incorporation of dopants. The brighter luminescence of the triangular defects, visible under UV light, is due to additional excitation. Different impurities or locally varying concentrations of standard impurities, as well as point defects accompanying ammonothermal and HVPE crystallization, may be responsible for this effect. It should be mentioned that changes in the level of impurities can be small and difficult to measure by, for example, secondary-ion mass spectrometry (SIMS). Performing PE on the area with triangular defects made it clear that these areas have a lower concentration of electrons. To clarify, the PE technique is extremely sensitive to small changes in the carrier concentration [21]. In addition, areas of its lower value are etched much faster in relation to those of a higher concentration [22,23]. In the case of the ammonothermal method, a lower electron concentration may be related to a local increase in compensating acceptors, such as Mg, Mn, or Fe [17]. On the other hand, a local decrease in the incorporation of electron donors, such as O and Si, can appear during crystal growth. This will be clarified in the course of further research.

The reason for the locally variable incorporation of impurities during crystal growth can certainly be a change in the growth mode. This is confirmed by the fact that areas of brighter luminescence under UV light coincide with step-bunching. In the course of the study, it was found that all parts where step-bunching was observed were associated with a specific alignment of the sides of two adjacent developed growth hillocks. Going into details, an obvious correlation was observed between the occurrence of step-bunching and the position of macro-steps on the sides of adjacent growth hillocks. Figure 8 shows three cases of merging macro-steps located on the slopes of two adjacent hillocks. Wherever the angle between macro-steps visible on neighboring sides of two hillocks is equal to 0° or 60°, no morphological disturbance is observed. These two situations are schematically illustrated in Figure 8a,b, respectively. When the macro-steps of adjacent hillocks form an angle of 120°, step-bunching disturbances are observed at the junction of the two sides (see Figure 8c). This is closely related to the anisotropic nature of GaN which manifests itself in different and direction-dependent growth rates under specific conditions [9,17]. A hillock reflects the hexagonal crystallographic structure of GaN and each of its six sides is inclined towards an *m* crystallographic direction. The edges of the steps on the hillock sides are parallel to the corresponding *m*-plane. The hexagonal shape of the hillock and the fact that steps are visible on its sides are a consequence of the anisotropy of GaN growth. This is evident in the crystallization of bulk crystals, where spontaneous formation of *m*-facets at the edge of the crystal is observed [9]. In addition, the anisotropy is evident in the difference of growth rate in the *a*- and *m*-crystallographic directions [15,17]. Thus, it can be concluded that the lowest-energy crystallographic planes in the mentioned growth conditions of the studied crystals are *m*-planes. Therefore, the *m*-steps (marked with dashed green lines in Figure 8) on the hillocks’ slopes are also low-energy steps. The propagation of stable *m*-steps is realized by the slow formation of step adatoms. Following that, an *m*-step can occur through a rapid addition of atoms to the resulting kink in the *a*-crystallographic direction.

When the steps on the sides of two adjacent growth hillocks are set parallel to each other, they form an angle equal to 0 degrees. In this case, the valley between the hillocks is filled by a stable expansion of *m*-steps in the *a*-direction. Such an example is illustrated in Figure 8a. The same stable filling of the valley between the sides of the hillocks is realized when the steps located on the slopes of two adjacent hillocks form a 60-degree angle. In such an arrangement, we can observe a regular meeting of the sides of the kinks in the cavity of the resulting angle or a parallel expansion in the *a*-direction of two steps on adjacent hillocks sides, as presented in Figure 8b. Figure 8c shows an example where the steps on two adjacent hillocks form an angle of 120 degrees. Based on the analysis of the hexagonal structure of GaN, it can be concluded that in such a relative position of growth hillocks, it is possible to form an unstable *a*-step at the junction of their sides. An expansion of newly formed *a*-steps is realized by a rapid addition of atoms to the step. A fast propagation of *a*-step in the *a*-direction is observed. Simultaneously, the expansion of the *a*-step in the *m*-direction is very slow. Due to this reason, the expansion is accompanied by the formation of stable *m*-steps and the formation of a characteristic “zigzag” macro-steps (see Figure 3b and Figure 4c) [24]. The mechanism of *a*-step formation and propagation in this case is similar to what can be observed during growth on a seed with a given off-cut to the *a*-direction [8]. It would seem that the formation of a stable and slow-growing *m*-step during the propagation of an *a*-step will cause it to slow down. This was observed after the formation of triangular terraces visible in the area with step-bunching growth mode in Figure 3b. It should be noted that growth in the *a*-direction is realized on steps located one above the other. Moreover, the slopes of the hillock are inclined at a certain angle with respect to the −c/c-plane. Therefore, growth is realized at multiple levels and there can be an overlap of propagated higher-level steps on lower-level steps. This effect can look like the sliding of an avalanche (see Figure 3b). A proposed explanation of this phenomenon is shown in Figure 9.

The starting point for explanation of this phenomenon is the fact that growth on the hillocks is realized at the highest growth rate. Steps on a hillock slope propagate through the movement of the kink around the centrum of the hill. At the junction of two hills between the hillock slopes for which steps form an angle equal to 120 degrees, an unstable *a*-step (see Figure 9a) could be created. Following that, its rapid expansion in the *a*-direction is observed, with the formation of stable *m*-steps on the sides of the terrace. As the hills grow, new *a*-steps may be created in the cavity of the formed terrace and steps on the hillock side, which is presented in Figure 9b,c. The formation of further steps in cavities, together with the growth of hillocks and the expansion of steps, leads to the buildup of terraces and the expansion of the step-bunching area (see Figure 9d).

The observed convergent relationships in the way the hillocks connect for the crystals from the two different methods allow us to conclude that the proposed model is universal. Regardless of the method of GaN crystallization and whether the growth is realized in the −c- or c-direction, the consequences of merging of adjacent hillocks remain similar in terms of presence of growth disturbances. Whenever the steps on the slope of one hillock merge with those on a neighboring hillock and form an angle of 120 degrees, a disturbance in growth morphology is observed. This is clearly visible in Figure 6a,c. According to the model, such a situation opens up the possibility of realizing growth in the *a*-direction. The consequence of the formation of an unstable *a*-step and its rapid propagation can be a change in the amount of unintentionally incorporated dopants or the number of intrinsic point defects [25]. The result is an apparent change in luminescence under UV illumination of areas with significant morphological disturbances. These areas, like those with visible disturbances for both Am-GaN and HVPE-GaN crystals, are characterized by brighter luminescence under UV light.

The above studies made it possible to clarify the nature of the formation of triangular defects, i.e., areas with a different concentration of carriers in volume. This is very important from the perspective of substrate preparation from a bulk crystal and subsequent epitaxy carried out on the obtained substrates (formation of stress or local step-bunching). In addition, the presence of electrical inhomogeneities in the substrate can affect the processing of the obtained structures. Thus, it is necessary to eliminate or reduce the formation of such disorders while conducting the growth process. The appearance of the presented defects depends on the method of conducting growth. In the case of the ammonothermal approach, where we are dealing with growth on dozens of crystals and the growth zone is very large, the supersaturation over a single crystal is uniform. Therefore, despite the growth of crystals several millimeters thick, the dominance of one hillock is not observed, but several are present, and growth is realized equivalently on them. As mentioned earlier, the formation of the first hillocks occurs at the restored −c-plane [9]. The formation of a dominant hillock in the ammonothermal method can, therefore, be realized by starting growth on a lenticular seed. In this case, the −c-surface will be reconstructed first in the center of the seed, where the off-cut angle is smallest. Then, in the initial stages of growth, the first hillock will begin to develop. After reaching an appropriate size, it will become the dominant hillock on which further growth will be realized [9]. In the case of crystallization in the *c*-direction from gas phase in HVPE reactors, forcing growth into a single hillock can be achieved directly by an intentional increase in supersaturation at a specified location on the seed. This is possible by adjusting the desired geometry of the growth zone, i.e., a proper positioning of precursor outlet nozzles or the use of appropriate metal elements that cause the decomposition of ammonia close to the edge of the seed [26]. In both cases, it is extremely desirable to carry out the fastest transition from the mode of growth on many hillocks to growth on a single one. 

## 5. Conclusions

In this paper, a detailed analysis of the morphology of GaN crystal growth obtained by the ammonothermal and HVPE methods was carried out. The research was conducted to determine the source of triangular planar defects in GaN substrates, visible in XRT as areas with locally different lattice parameters. An analysis of the impact of mutual arrangement of growth hillocks was performed. It was found that in the area of the studied defect, the concentration of carriers is lower than in the rest of the crystal. In addition, the formation of such areas was associated with a change in the growth morphology. The latter was, in turn, attributed to the interconnection of adjacent growth hillocks. Three ways of connecting the steps flowing from the slope of two adjacent hillocks were analyzed. When the *m*-steps on two adjacent hillocks form angles of 0 and 60 degrees, the valley between the hillocks is filled by a stable expansion of *m*-steps in the *a*-direction. If this angle is equal to 120 degrees, an unstable *a*-step is formed. Its expansion is realized at a higher speed than the expansion of the *m*-step. This consequently leads to the formation of growth disturbance, which may result in the formation of inhomogeneous areas in the volume of the growing crystal. Ways of guiding the process to allow a rapid transition to growth on a single hillock were proposed.

## Figures and Tables

**Figure 1 materials-16-03360-f001:**
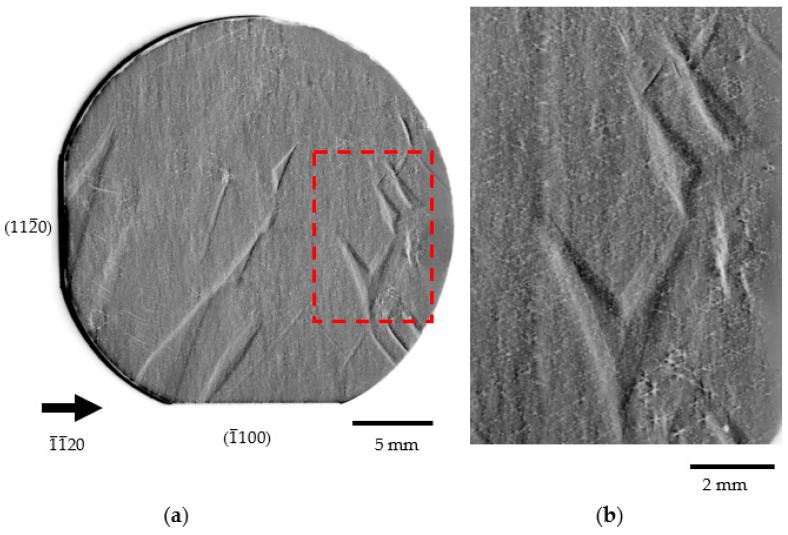
X-ray topography performed in transmission mode showing the Borrmann effect performed on 1-inch Am-GaN n-type substrate. View on: (**a**) full substrate with triangular planar defects, and (**b**) enlarged image of the dashed rectangle indicated area of Figure 1a showing triangular defects.

**Figure 2 materials-16-03360-f002:**
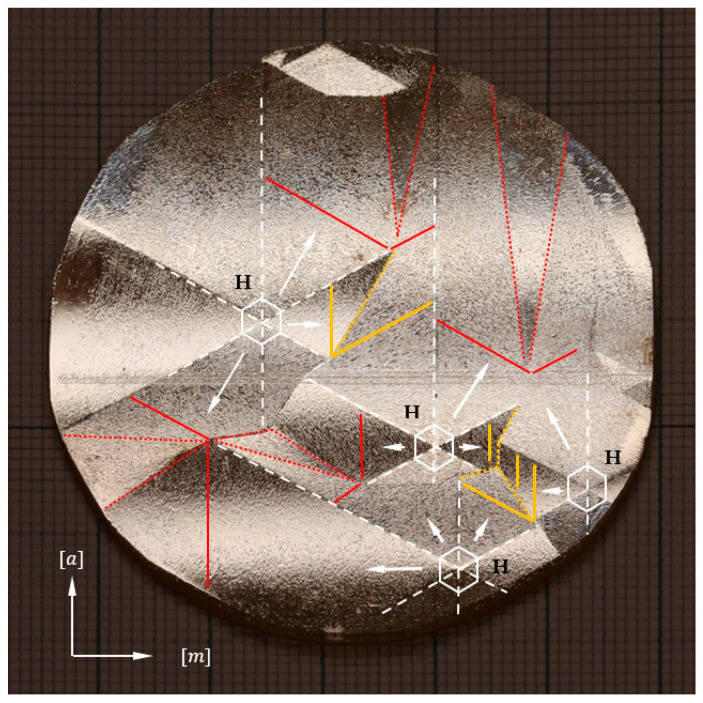
Morphology of 2-inch as-grown Am-GaN crystal. Indications: H—developed hillocks; solid white lines—macro-steps on the sides of hillocks; white dashed lines—hillock edges; solid yellow lines—macro-steps non-adjacent to step-bunching morphology; dotted yellow lines—mutual connection of hillocks without disturbances; solid red lines—macro-steps adjacent to area with step-bunching morphology; dotted red lines—areas with step-bunching morphology.

**Figure 3 materials-16-03360-f003:**
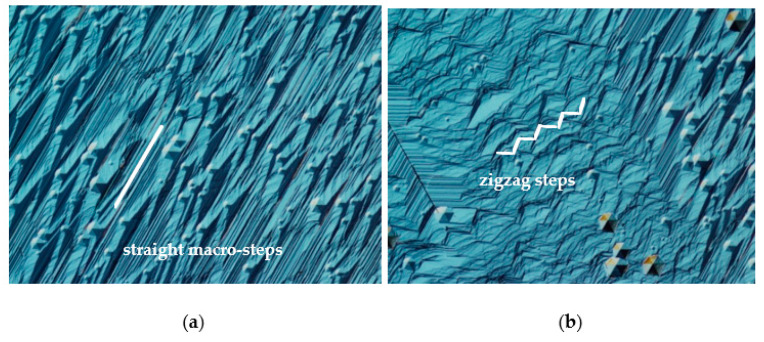
Images of the growth morphology: (**a**) straight macro-steps; (**b**) step-bunching, zigzag steps. Hexagonal pits are visible; they are the result of etching the surface of the crystal after the growth process to remove the polycrystalline layer formed during the reactor cooling stage.

**Figure 4 materials-16-03360-f004:**
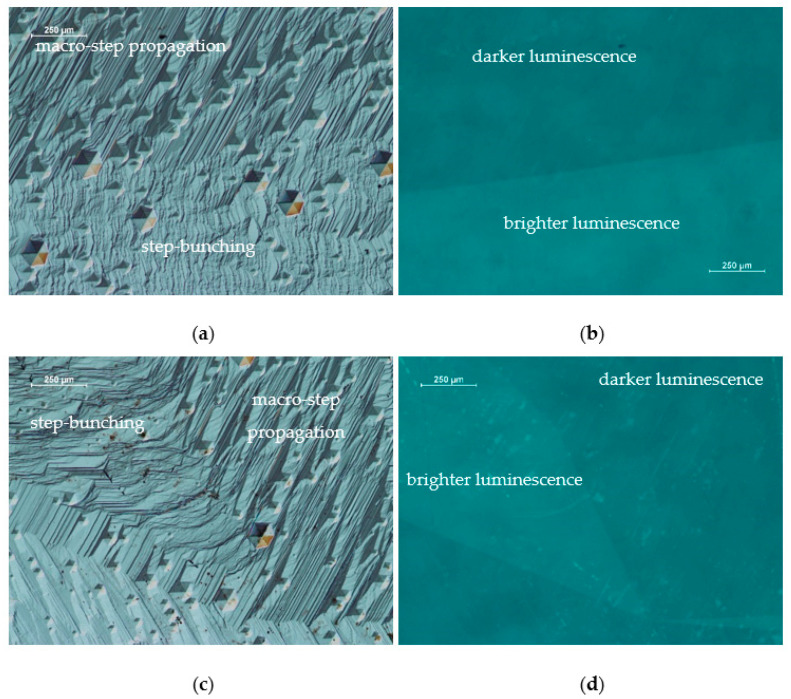
Images from OM of the Am-GaN crystal at the junction of two different growth morphologies performed: (**a**,**c**) with Nomarski contrast; (**b**,**d**) the same areas as Figure 4a,c, respectively, under UV light. All areas where step-bunching correspond to areas of brighter luminescence.

**Figure 5 materials-16-03360-f005:**
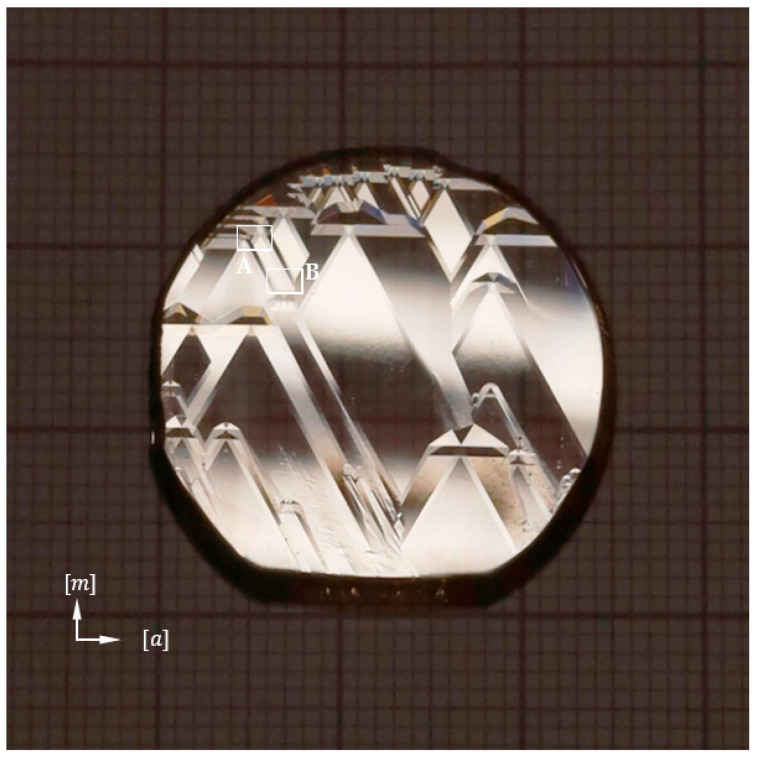
Image of the c plane morphology of 1-inch as-grown HVPE-GaN crystal; solid white rectangles A and B are areas that are analyzed in Figure 6.

**Figure 6 materials-16-03360-f006:**
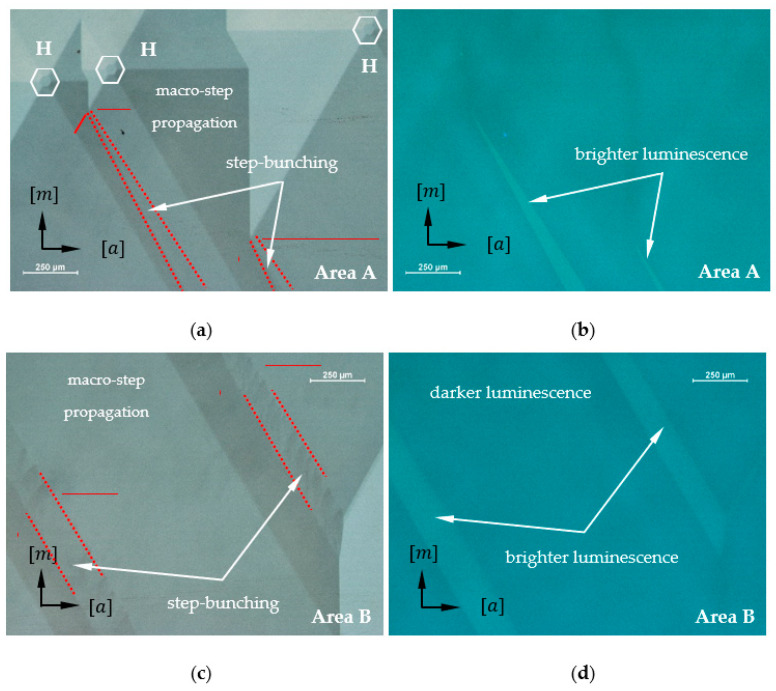
Images of the HVPE-GaN crystal performed by OM at areas indicated in Figure 6: (**a**,**c**) growth morphologies performed with Nomarski contrast; (**b**,**d**) the same areas as Figure 6a,c, respectively, under UV illumination. Red solid and dotted lines mark, respectively, steps on the sides of the hillocks (indicated as H), and areas where step-bunching is observed.

**Figure 7 materials-16-03360-f007:**
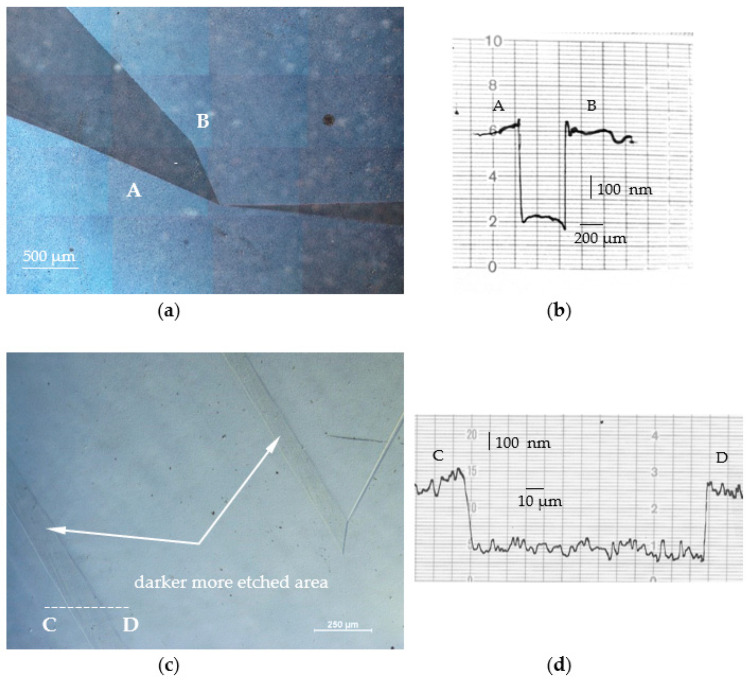
Optical microscope image with Nomarski contrast of the surface after photo etching of: (**a**) Am-GaN crystal; (**c**) HVPE-GaN crystal. Dashed white lines indicate segments AB and CD where profilometry was performed. Diagram of the change in depth on segment: (**b**) AB and (**d**) CD.

**Figure 8 materials-16-03360-f008:**
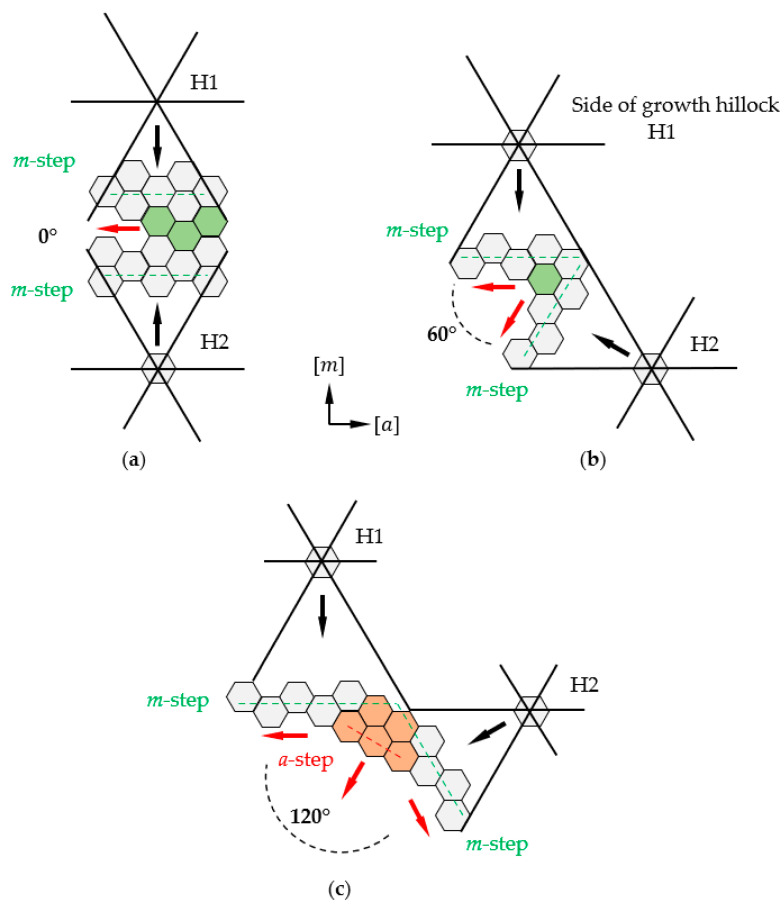
Illustration of ways of connection of two adjacent growth hillocks (indicated as H1 and H2). Dependence of the occurrence of step-bunching on the angle formed by macro-steps on the sides of two adjacent growth hillocks. Angle equal to: (**a**) 0 degrees—no disturbance at the junction of macro-steps; (**b**) 60 degrees—no disturbance at the junction of macro-steps; (**c**) 120 degrees—step-bunching present at the junction of macro-stapes. Black arrows indicate the direction of *m*-step propagation on the growth hillocks. The red arrows indicate the directions of rapid reconstruction of m-step and directions of *a*-step propagation.

**Figure 9 materials-16-03360-f009:**
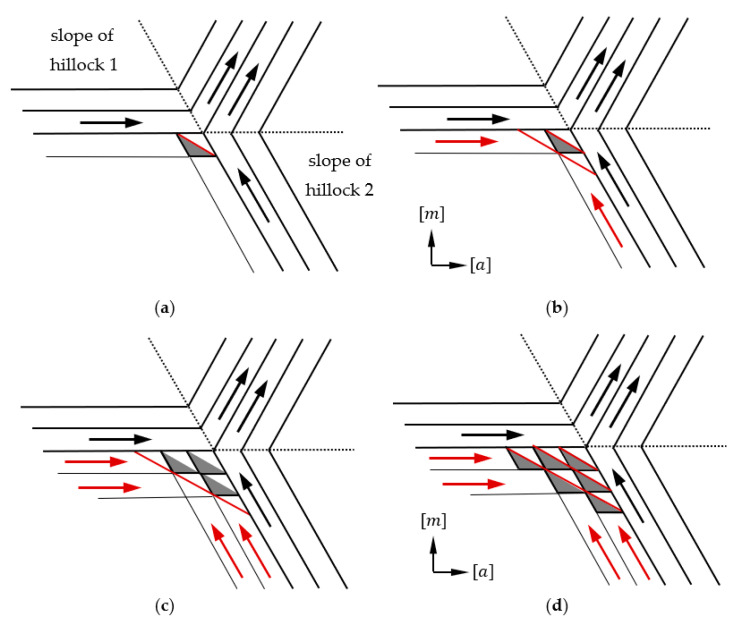
Illustration of the propagation of unstable *a*-steps at the junction of two slopes of adjacent hillocks which form an angle of 120 degrees: (**a**) formation of an unstable *a*-step and its rapid expansion in *a*-direction; arrows indicate the direction of expansion of the *m*-step on the hillock’s slope; (**b**) formation of an unstable *a*-step at the junction of the edges of the triangular terrace and the hillocks’ slopes; (**c**) formation of new terraces as a result of the propagation of the *a*-steps and new unstable *a*-steps between the terraces; (**d**) expansion of newly created *a*-steps.

## Data Availability

Not applicable.

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
