# Peer review of "Evolution of the Growth Mode and Its Consequences during Bulk Crystallization of GaN"

_materials, 2023, doi:10.3390/ma16093360_

Round 1
Reviewer 1 Report
Paper is written well. It can be accepted in its present form.
Author Response
Dear Reviewer
Thank you very much for your positive opinion about our publication.
Reviewer 2 Report
This paper has investigated the triangle planar defects formed in GaN crystals formed both by the ammonothermal and HVPE methods. The authors suggested a comprehensive model of the interconnection of adjacent growth hillocks. If the angle of the m-steps on two adjacent hillocks is 120 degrees, unstable a-steps which are the origin for the zigzag morphology forms between them. A fast propagation of the a-steps results the difference in the impurity concentration (carrier concentration) in the step-bunched zigzag areas. The authors confirmed that the carrier concentration in the step-bunching area is lower than that in the rest of the crystal based on the difference in the PE rate sensitive to the electronic concentration.
This paper is well-written and effectively organized, and the focus of this study is well phrased in the Introduction. The study provides an important contribution to the crystal growth society. The reviewer recommends this paper for acceptance for publication in Materials.
I have just a few comments, which would be helpful to improve the clarity of the paper.
P11, L263-268
The authors discussed the relationship between the formation of unstable a-steps and incorporated dopant concentration. This is a very interesting point in crystal growth, especially for growth from solution in which the macrostep formation is significant. Is it possible to add more considerations on the relationship between the step propagation speed and the incorporation ratio of the impurity elements with a segregation coefficient less than unity.
P11, L250-254
The authors explained the bunching tendency on the slope where unstable a-steps appear, in a such way that the higher-level a-steps overlap on lower-level steps. In this case the reviewer thinks that the formation of vast c-planes due to the overlapping of steps would be manifested under this mechanism. However, experimental results show that the surface roughness of the area seems to be comparable with other areas. If possible, please add some comments on what the determining factor on the degree of the step-bunching (overlapping of steps) is.
Author Response
Dear Reviewer
Thank you very much for your review of our article. I am sending the answers to your questions and suggestions in the attached file.

Reviewer 3 Report
In this manuscript, the authors report research of morphology of gallium nitride crystal growth obtained by ammonothermal and HVPE methods. The source of triangular planar defects in GaN substrates, visible in XRT as areas with locally different lattice parameters were determined by the morphology research. However, the authors should clarify and carefully consider the following comments and I would suggest its publication after major revisions.
1. The authors mentioned that the as-grown surfaces which for the ammonothermal and HVPE methods correspond to −?- and ?-surfaces, respectively. Bulk crystallization in polar −? and ? directions on seeds with a given misorientation can be realized in a similar way. However, the –? surfaces are N-polar faces, and the ? surfaces are Ga polar faces. The surface atom structure and properties are different. The authors may provide a more reasonable explanation about this issue.
2. Optical microscope image with Nomarski contrast of the surface after photo etching were shown in Figure 7. These crystals were mechanical and chemo-mechanical polished, cleaned and photo etched. Why did authors carry out the above processing steps? What about the optical microscope image result of GaN crystal as-grown surface? The influence and purpose of the processing and PE steps should be given.
3. In Figure 5, image of the ? plane morphology of 1-inch as-grown HVPE-GaN crystal was shown. However, the size of the crystal cannot be identified and the grid cannot be seen. Meanwhile, the shooting direction seems to not be c direction like in Figure 2. It leads to the 1-inch crystal not being circular.
Author Response

(The authors gave the same response as above.)

Round 2
Reviewer 3 Report
The authors have replied to our comments and have made the changes in the article. Now I recommend this paper to be published.